# Mercury Removal from Aqueous Solution Using ETS-4 in the Presence of Cations of Distinct Sizes

**DOI:** 10.3390/ma14010011

**Published:** 2020-12-22

**Authors:** Simão P. Cardoso, Tiago L. Faria, Eduarda Pereira, Inês Portugal, Cláudia B. Lopes, Carlos M. Silva

**Affiliations:** 1CICECO, Department of Chemistry, University of Aveiro, Campus de Santiago, 3810-193 Aveiro, Portugal; simaocardoso@ua.pt (S.P.C.); tiago.luis@live.ua.pt (T.L.F.); inesport@ua.pt (I.P.); 2QOPNA & LAQV-REQUIMTE, Department of Chemistry, University of Aveiro, 3810-193 Aveiro, Portugal; eduper@ua.pt

**Keywords:** ETS-4, ion exchange, kinetics, mercury, tetrapropylammonium hydroxide

## Abstract

The removal of the hazardous Hg^2+^ from aqueous solutions was studied by ion exchange using titanosilicate in sodium form (Na-ETS-4). Isothermal batch experiments at fixed pH were performed to measure equilibrium and kinetic data, considering two very distinct situations to assess the influence of competition effects: (i) the counter ions initially in solution are Na^+^ and Hg^2+^ (both are exchangeable); (ii) the initial counter ions in solution are tetrapropylammonium (TPA^+^) and Hg^2+^ (only Hg^2+^ is exchangeable, since TPA^+^ is larger than the ETS-4 micropores). The results confirmed that ETS-4 is highly selective for Hg^2+^, with more than 90% of the mercury being exchanged from the fluid phase. The final equilibrium attained under the presence of TPA^+^ or Na^+^ in solution was very similar, however, the Hg^2+^/Na^+^/ETS-4 system in the presence of Na^+^ required more 100 h to reach equilibrium than in the presence of TPA^+^. The Hg^2+^/Na^+^/ETS-4 system was modelled and analyzed in terms of equilibrium (mass action law) and mass transfer (Maxwell–Stefan (MS) formalism). Concerning equilibrium, no major deviations from ideality were found in the range of studied concentrations. On the other hand, the MS based model described successfully (average deviation of 5.81%) all kinetic curves of mercury removal.

## 1. Introduction

Water pollution is one of the main concerns the World is currently facing. The continuous increase of the world population, the need for high quality drinking water, and water scarcity in many regions of our planet make water treatment processes a hot subject among researchers.

Heavy metals are considered traditional water contaminants. As a result of many applications in industrial, construction and agricultural sectors, there is inevitably some loss and dispersion of metals back into the environment [1]. In addition to the so-called anthropogenic sources, metal releases can also be of natural origin, like rocks and soil erosion, emissions from volcanoes, and atmospheric deposition [1]. Mercury is one of the most hazardous metal contaminants, since its presence in water, even at very low concentrations, causes severe harmful effects to biota. This happens because of bioaccumulation and biomagnification [2], which means that the concentration of Hg^2+^ in the biota is much higher than in water. For these reasons mercury was classified as a priority pollutant by the European Union [3] and is in the Top 3 of the priority list of hazardous substances established by the Agency for Toxic Substances and Disease Registry [4].

Nowadays, with the legal restrictions imposed in the so-called developed countries, mercury contamination cannot be solely linked to localized individual emissions of the metal being more associated to the widespread air pollution instead [5]. Consequently, in remote areas free of anthropogenic pressure the mercury levels found in fish are very high despite its extraordinarily low levels in air and surface water [5]. This fact raises a new question regarding water treatment. Conventional and widely used processes like precipitation, that is very effective for high metals concentrations, may not be efficient to remove the current levels of mercury found in water. In this context, the use of high capacity and selective sorbent materials (e.g., ion exchangers and adsorbents) can be the right solution for the removal of low levels of contamination. Materials like zeolites, clays, carbons, polymers, and biosorbents from agricultural and agro-forest residues, have been extensively studied in the literature [1,6,7,8,9,10,11,12,13,14,15,16,17,18,19,20,21,22,23,24].

Synthetic microporous materials like titanosilicates and zirconosilicates have been successfully studied as ion exchangers for divalent metals uptake [1,6,7,8,9,10,11,12,13,14,15,16]. These materials have attracted considerable attention during the last decades since they are stable, possess a narrow pore size distribution, consist of a variety of framework structures, exhibit high capacity and frequently a remarkable selectivity for target species. The selectivity of these materials is attributed to the geometric matching of their channels and/or cavities with the size of the counter ions, which promotes interatomic/ionic interaction and coordination [25]. Moreover, it is well-known that these materials have a negatively-charged skeleton balanced by the presence of exchangeable cations held electrostatically within the micropores. These cations (frequently Na^+^ or K^+^) may play an important role on the extension and kinetics of the process, since their size and/or coordination forces can enhance or hinder the ion exchange [26].

ETS-4 (Engelhard Titanium Silicates No. 4) presents one of the highest ion exchange capacity among all reported titanosilicate materials (theoretical capacity of 6.39 eq kg^−1^), which makes it an excellent ion exchanger within a limited temperature range (<200 °C) [27,28]. However, few studies exist in the literature involving this solid: Al-Attar and Dyer [29] studied the sorption of uranium onto titanosilicate materials, among them ETS-4; Popa et al. [30] applied ETS-4 for the purification of waste waters containing radioactive ions (^60^Co^2+^, ^115m^Cd^2+^, and ^203^Hg^2+^), and later published a review on titanosilicates materials for radioactive wastewaters purification [31]; Lopes et al. [32,33] and Ferreira et al. [11] used ETS-4 for Hg^2+^ and Cd^2+^ removal from water, respectively, and later Cardoso et al. [12] studied the competitive removal of these cations (Cd^2+^ and Hg^2+^) from water ; Figueiredo et al. [15] investigated the removal of Cs^+^ from aqueous solutions carrying out both batch and fixed-bed experiments with ETS-4.

Lopes et al. [34] studied the impact of pH and temperature on the ETS-4 uptake efficiency of Hg^2+^ from solution and concluded that its removal increases with increasing pH, until a maximum value around pH 4–6, and with decreasing temperature. These conditions are very attractive from the industrial point of view since the pH of domestic, medical, and industrial effluents is generally in this interval and, hence, no significant pH adjustments are necessary [13,35,36,37]. To test the capacity of ETS-4 in these conditions, NaOH is commonly added to adjust the system’s pH. One may anticipate that the addition of Na^+^ cations to the solution can influence the mercury removal efficiency, since they enter in the porous structure of ETS-4 and compete with Hg^2+^ ions for the active site. In this essay, in order to analyze this competition rigorously, a tetrapropylammonium hydroxide solution (TPAOH) is chosen as pH adjuster because its cation (tetrapropylammonium cation, TPA^+^) is larger than the pores of ETS-4 and thus cannot penetrate the solid.

The present work focuses the experimental and modelling study of the equilibrium and kinetics of the Hg^2+^/Na^+^/ETS-4 system, using a TPAOH solution as pH adjuster. In this way, the stringent competition between the large concentration of Na^+^ in solution (when NaOH is used) and the target Hg^2+^ cations is rigorously analyzed in terms of equilibrium and mass transfer. The mass action law, expressed in terms of activities, is used to model the equilibrium, and the Ioannidis et al. method [38,39] is adopted to optimize noncorrelated parameters. With respect to kinetics, the Maxwell–Stefan (MS) formalism is chosen and compared with semi-empirical pseudo-first and pseudo-second order equations.

## 2. Materials and Methods

### 2.1. Chemicals

All reagents used in this work were of analytical grade and used without further purification. Sodium hydroxide (NaOH) 0.1 M, potassium chloride (KCl), nitric acid (25 vol%) and tetrapropylammonium hydroxide solution (TPAOH, $1.0\times 10^{-3}$ mol m^−3^) were purchased from Merck (New York, NY, USA). The certified standard solution of Hg(NO_3_)_2_ (1000 ± 1 kg m^−3^) was purchased from Spectrosol^®^ BDH. All working solutions, including standards for the calibration curves, were obtained by diluting the corresponding stock solutions in ultra-pure water (Millipore, Integral 10 system, Molsheim, France).

The TPAOH solution was analyzed by cold vapor-atomic fluorescence spectrometry (hydride/vapor generator PS Analytical Model 10.003 (PS Analytical, Kent, UK), coupled to a PS Analytical Model 10.023 Merlin atomic fluorescence spectrometer (PS Analytical, Kent, UK) in order to quantify its original amount of mercury.

### 2.2. Engelhard Titanium Silicate No. 4 (ETS-4)

ETS-4 is the synthetic analogue of the mineral zorite [38] with chemical composition [Na_9_Ti_5_Si_12_O_38_(OH)·4H_2_O]. It is one of the main members of a class of heteropolyhedra transition metal silicates with pore size of $\left(30-40\right)\times 10^{-9}$ m and particle size of $\left(500-900\right)\times 10^{-9}$ m [10]. Briefly, the synthesis [33] started with the preparation of an alkaline solution, by dissolving metasilicate (BDH), NaOH and KCl in H_2_O. Then, TiCl_3_ (15 wt % TiCl_3_ and 10 wt % HCl, Merck) was added to this solution and stirred thoroughly. The gel, with molar composition 5.9 Na_2_O:0.7 K_2_O:5.0 SiO_2_:1.0 TiO_2_:114 H_2_O, was then transferred to a Teflon-lined autoclave and treated at 230 °C for 17 h under autogenous pressure without agitation. The product was filtered off, washed at room temperature with distilled water, and dried at 70 °C overnight.

The framework of this microporous material comprises corner-sharing SiO_4_ tetrahedra, TiO_5_ pentahedra and TiO_6_ octahedra, and each titanium ion has an associated (−2) charge, which is neutralized by extra-framework cations, usually Na^+^ [33]. The possibility of replacement of those cations by others like Hg^2+^ makes ETS-4 a good ion exchange material.

### 2.3. Ion Exchange Studies and Analytical Procedures

Prior to use, all glassware necessary for the ion exchange studies was acid washed with HNO_3_ (25 vol%), during 24 h, to inactivate the glass surface and then rinsed abundantly with ultra-pure water.

The ion exchange experiments were carried out under isothermal batch conditions (295 ± 1 K) in 2 $\times 10^{-3}$ m^3^ volumetric flasks, closed to avoid losses of mercury. The starting time of each experiment was the instant of ETS-4 addition to the solution. The suspensions were magnetically stirred (1400 rpm) and aliquots were collected at increasing times, using a $10\times 10^{-6}$ m^3^ syringe coupled to a Teflon tube, filtered through a Millipore membrane with pore size of $0.45\times 10^{-6}$ m (previously washed with nitric acid solution (2 vol%)), adjusted to pH < 2 with concentrated HNO_3_, and then analyzed by cold vapor-atomic fluorescence (as described in Section 2.1) to ascertain the mercury concentration.

#### 2.3.1. Ion Exchange of TPA^+^ Solution (without Hg^2+^) Using Na-ETS-4

Rigorous masses of ETS-4 (pristine and washed with ultra-pure water to remove any vestigial amounts of Na^+^ from its synthesis) were added to ultra-pure water containing $0.5\times 10^{-6}$ m^3^ of TPAOH $0.1\times 10^{-3}$ mol m^−3^ (doses of 7.71 and 8.33 g m^−3^). Triplicate aliquots were collected (i) before the addition of TPAOH and ETS-4, (ii) immediately after the addition of TPAOH, just before the addition of ETS-4, (iii) immediately after the addition of ETS-4, and (iv) after three and seven days of contact. The Na^+^ concentration of the aliquots was measured by flame atomic absorption spectroscopy (Perkin Elmer, Überlingen, Germany), after filtering.

#### 2.3.2. Ion Exchange of Hg^2+^ Solution Using Na-ETS-4

The initial Hg^2+^ concentration was fixed in all assays (ca. $5\times 10^{-3}$ mol m^−3^; see Table 1) and the solutions were prepared by diluting the mercury stock solution in ultra-pure water (18.2 MΩ cm). The solution pH was adjusted to 6 [34] using NaOH or TPAOH solutions. For the kinetic studies, rigorous masses of pre-washed ETS-4 (doses of 13.64, 15.63 and 15.09 g m^−3^, denoted by Expression (6), Expression (9), and Expression (14), respectively) were added to the mercury solution. In Expressions (6) and (9) the pH was adjusted with TPAOH while in Expression (14) it was adjusted with NaOH. For quality control, a blank experiment (i.e., without ETS-4) was always run in parallel under the same operating conditions. Duplicate aliquots were collected at 0, 15, 30, and 45 min, and at 1, 1.5, 2, 4, 8, 20, 30, 45, 70, 100, and 200 h. The horizontal branch of each curve was used to determine the equilibrium isotherm. Additional experiments were carried out by changing the dose of ETS-4 (7.15–25.85 g m^−3^; see Table 1) to obtain new isotherm data points, for which only the initial and equilibrium concentrations were measured (0, 24, 48, 72, and 80 h). After solid-solution separation, the Hg^2+^ was quantified by cold vapour-atomic fluorescence (as mentioned in Section 2.1) with a range of standards for calibration between $\left(0.1\;\mathrm{and}\;0.5\right)\times 10^{-3}$ g m^−3^, a limit of detection of $0.3\times 10^{-4}$ g m^−3^ and a precision and accuracy < 5%. Several blanks and standards were always analyzed with the sample batch.

The average Hg^2+^ concentration in the solid at time t, $\langle q_{A}\rangle$ (mol m^−3^) was calculated by material balance according to:(1)$$\langle q_{A}\left(t\right)\rangle =\frac{V_{L}}{m_{s}}\left[C_{A,0}-C_{A}\left(t\right)\right]\times \rho_{s}$$
where subscript A denotes Hg^2+^, $C_{A,0}$ (mol m^−3^) is the initial mercury concentration in solution, $C_{A}\left(t\right)$ is the mercury concentration in solution at time t, $V_{L}$ (m^3^) is the volume of solution, and $m_{s}$ (kg) and $\rho_{s}=2200$ kg m^−3^ are the mass and the density of ETS-4, respectively. The percentage of Hg^2+^ removed was calculated by:(2)$$R\left(t\right)=100\times \frac{C_{A,0}-C_{A}\left(t\right)}{C_{A,0}-C_{A,e}}$$
where $C_{A,e}$ is the mercury concentration in solution at equilibrium.

## 3. Modelling

The ion exchange process was accurately modelled and analyzed in terms of equilibrium and kinetics. Concerning equilibrium, the mass action law was combined with Debye–Hückel equation and Wilson model to obtain the activity coefficients in the fluid and exchanger phases, respectively (see Appendix A) [13,39,40]. The Maxwell–Stefan (MS) phenomenological model was adopted to describe the ion exchange mass transfer [41,42]. The MS formalism, the material balances, the initial and the boundary conditions, and the numerical solution approach are described in the Appendix A. For comparison, the pseudo-first and pseudo-second order rate equations (see Appendix A) were also adopted.

The MS diffusivities, the convective mass transfer coefficient and the sorption rate constants of the semi-empirical equations were optimized by fitting the appropriate models to the experimental data. The MS model was solved numerically using the Method of Lines and a finite-difference approach. The Nelder–Mead algorithm was used to optimize the parameters, minimizing the average absolute relative deviation (AARD) defined by:
(3)$$\mathrm{AARD}\left(\%\right)=\frac{100}{\mathrm{NDP}}\overset{\mathrm{NDP}}{\underset{i=1}{\sum }}\left|\frac{C_{A,i}^{\mathrm{calc}}-C_{A,i}^{\exp}}{C_{A,i}^{\exp}}\right|$$
where NDP is the number of data points, and superscripts calc and exp refer to the calculated and experimental mercury concentration in the solution. Matlab R2014a^®^ software was used for all calculations.

## 4. Results and Discussion

### 4.1. Check that Tetrapropylammonium Cation (TPA^+^) Cannot Penetrate ETS-4 Pores

In this study a TPAOH solution was selected for pH adjustments. Since the mercury concentration in this TPAOH solution was only $3.94\times 10^{-5}$ mol m^−3^, which represents less than 0.8% of the initial mercury concentration ($C_{A,0},\mathrm{ca}.\;5\times 10^{-3}{\mathrm{mol}\;m}^{-3}$), this contribution was considered irrelevant. The values of $C_{A0}$ were determined analytically for all the assays.

The possibility of exchanging the Na^+^ initially present in the ETS-4 by the TPA^+^ used to adjust the solution pH was investigated. Results showed that the ultra-pure water before and after the addition of the TPAOH solution was free of sodium (<0.01 g m^−3^). Nonetheless, immediately after the addition of ETS-4 the sodium concentration increased and remained constant after three and seven days, with an increment of $\left(4.7\pm 0.4\right)\times 10^{-5}$ g of Na^+^ per gram of ETS-4 and liter of solution. The same approach was carried out with ETS-4 previously washed with ultra-pure water, aiming to remove possible trace amounts of Na^+^ coming from its synthesis. Results showed that when pre-washed ETS-4 was used the solution was free of sodium (<0.01 g m^−3^), even seven days after the ETS-4 addition to the TPAOH solution. This confirms that TPAOH solution is not a source of Na^+^ to the working solution and that the TPA^+^ is large enough to penetrate into the crystal’s pores and exchange with the Na^+^ of the sorbent. In conclusion, the presence of the Na^+^ cation in solution in the first assay came from the ETS-4 surface and not from the pores of its framework.

### 4.2. Isotherm of the Hg^2+^/Na^+^/ETS-4 System

Figure 1 displays the isotherm of the Hg^2+^/Na^+^/ETS-4 system expressed as molar concentration of mercury in the solid ($q_{A}$) as a function of the equilibrium concentration in solution ($C_{A,e}$). The isotherm is apparently linear in all range of studied concentrations.

Concerning modelling, the Ioannidis et al. [39,43] approach was adopted to obtain the equilibrium constant, $K_{B}^{A}$, (where A is Hg^2+^ and B is Na^+^—see Appendix A) for two combinations of the solid activity coefficient (${\bar{\gamma }}_{i}$)—specifically, the Wilson and NRTL models—and Debye–Hückel for the fluid phase activity model ($\gamma_{i}$). The adjusted $K_{B}^{A}$, the Wilson parameters ($\Lambda_{12}$ and $\Lambda_{21}$) and the NRTL parameters ($g_{12}$ and $g_{21}$) are listed in Table 2 together with the calculated AARD. From Figure 1 and Table 2 it is possible to observe the goodness of fit using the activity coefficient models (AARD = 0.89%) comparatively to the ideal situation (AARD = 0.93%) with the ideal isotherm overpredicting the results. Figure 1 also shows that the equilibrium curves obtained using the activity coefficient models are essentially overlapped, regardless of the chosen model for the solid phase, resulting in similar $K_{B}^{A}$ for both models. On other hand, the small ionic strength of the solutions, $I=\left(6.74-6.83\right)\times 10^{-4}$ molal, contributes for the small deviations from ideality, allowing a reasonable adjustment of the ideal model to the results.

### 4.3. Kinetics Modelling of the Hg^2+^/Na^+^/ETS-4 System

The experimental data for the normalized Hg^2+^ concentration in the fluid and in the solid along time are plotted in Figure 2 together with the MS modelling results. The experiments differ in the solution used for pH adjustment (namely, TPAOH solution and NaOH solution), and in the ETS-4 load (namely, 13.64 and 15.09/15.63 g m^−3^).

The results suggest that independently of the pH adjusting solution, the addition of a few milligrams per litre of ETS-4 significantly decreases the Hg^2+^ concentration in solution (Figure 2a). As expected, the ion exchange of Hg^2+^ is faster at the beginning, slowing down until equilibrium attainment. Such kinetic pattern is due to the large mass transport driving forces that prevail at the beginning of the ion exchange process, since ETS-4 particles are initially free of metal ions. However, depending of the pH adjusting solution (TPAOH or NaOH) relevant differences were observed on the kinetics of Hg^2+^ uptake, as the final equilibrium is attained much faster when using TPAOH than when using NaOH solution. In fact, the Hg^2+^/Na^+^/ETS-4 system in the presence of NaOH solution needed more 100 h to reach equilibrium than in the presence of TPAOH solution, since the NaOH solution reduces significantly the driving force for mass transport of sodium and hence the ion exchange process is slower than when using the TPAOH solution. This statement is corroborated mathematically by the initial uptake rates, which were estimated from the first derivative of $C_{A}=f\left(t\right)$ at t=0. The evolution of the solid loading, $\langle q_{A}\rangle \left(t\right)$, in Figure 2b complements and confirms these interpretations, since $\langle q_{A}\rangle \left(t\right)$ and $C_{A}\left(t\right)$ are linearly related by Equation (1).

The initial uptake rates of Hg^2+^ when using TPAOH solution were $1.50\times 10^{-3}$ and $1.69\times 10^{-3}$ mol m^−3^ h^−1^ for 13.6 and 15.6 g m^−3^ of ETS-4, respectively. When the pH was adjusted with a NaOH solution the initial uptake rate of Hg^2+^ was $1.0\times 10^{-3}$ mol m^−3^ h^−1^, (with 15.1 g m^−3^ of ETS-4). The kinetic selectivity (i.e., the ratio between the removal rates when using TPAOH and NaOH solutions, for ca. 15 g m^−3^ of ETS-4) was 1.7, confirming that the kinetics of Hg^2+^ exchange is slower when Na^+^ is added to the working solution. These results can be explained by the competition effect between Na^+^ and Hg^2+^ in the ion exchange phenomenon, with the concentration of Na^+^ being much higher than the concentration of Hg^2+^ (when the pH is adjusted with the NaOH solution). In fact, comparing Exps. 9 and 14 (which use TPAOH and NaOH, respectively), the initial concentration of Na^+^ in solution was 0 and $7.55\times 10^{-1}$ mol m^−3^, respectively, whereas the initial Hg^2+^ concentration was $5.13\times 10^{-3}$ and $5.43\times 10^{-3}$ mol m^−3^, respectively.

As equilibrium is approached, the normalized Hg^2+^ concentration in the fluid converges to the same value ($C_{A}/C_{A0}=$ 0.13 ± 0.01) regardless of the pH adjusting solution At equilibrium, and depending on the amount of ETS-4, 86 to 88% of Hg^2+^ was exchanged by the Na^+^ initially present in the solid. These results clearly demonstrate that ETS-4 is highly selective for Hg^2+^ cation and at equilibrium the ion exchange process is not affected by the presence of high amounts of Na^+^ in solution (which is the case when a NaOH solution is used for pH adjustment).

A considerable improvement on the ion exchange results of Hg^2+^ was achieved by increasing the amount of ETS-4. For example, independently of the pH adjusting solution, duplicating the ETS-4 dose increased Hg^2+^ removal from 84.5% (ca. 12.56 g m^−3^ of ETS-4, Exps. 4 and 13) to 92.5% (ca. 25.52 g m^−3^, Exp. 12 and 15) (Figure 3). The extensive ion exchange capacity is proportional to the solid mass since the initial mercury concentration in solution was fixed around 1 g m^−3^ in all assays. The equilibrium selectivity (S) was calculated by the ratio between the distribution coefficients of Hg^2+^ at equilibrium, when using TPAOH and NaOH solutions ($S=\left(q_{A,\mathrm{eq}}^{\mathrm{TPAOH}}/C_{A,\mathrm{eq}}^{\mathrm{TPAOH}}\right)/\left(q_{A,\mathrm{eq}}^{\mathrm{NaOH}}/C_{A,\mathrm{eq}}^{\mathrm{NaOH}}\right)$). The value found for the equilibrium selectivity was 1.02 ± 0.04, confirming that the presence of Na^+^ cation in solution does not mitigate the extension of the ion exchange process.

With respect to the results obtained with the MS model (lines in Figure 2) the good agreement between modelling and measurements is noteworthy, corresponding to AARD = 5.81% (see Table 3). A trustworthy correlation is accomplished even in the transition region from the steep descent to the horizontal branch, where the kinetic curves are usually very difficult to fit.

The MS diffusion coefficients (Table 3) are in the order of 10^−16^–10^−21^ m^2^ s^−1^, being consistent with the small pore diameters of ETS-4 ((30–40) × 10^−9^ m) [44] and similar to literature values for other systems (Table 4). For instance, similar values can be found in the following articles: (i) Barrer and Rees [45] reported self-diffusion coefficients of $1.14\times 10^{-17}$, $1.96\times 10^{-21}$, and $8.27\times 10^{-19}$ m^2^ s^−1^ for Na^+^, K^+^, and Rb^+^ in analcite, respectively; (ii) Coker and Rees [46] obtained apparent diffusion coefficients of $1.8\times 10^{-17}$ and $8.0\times 10^{-18}$ m^2^ s^−1^ for Ca^2+^ and Mg^2+^ in semi-crystalline Zeolite Na-A, respectively; and (iii) Lopes et al. [47] obtained diffusion coefficients of $1.108\times 10^{-19}$ and $7.873\times 10^{-19}$ m^2^ s^−1^ for Hg^2+^ and Na^+^ in Hg^2+^/Na^+^/ETS-4 system, respectively, using the Nerst-Plank model.

Concerning the convective mass transfer coefficient, the fitted value ($5.09\times 10^{-4}$ m s^−1^) differs from that predicted by the Armanante and Kirwan correlation [48] ($2.90\times 10^{-3}$ m s^−1^). However, this method is not entirely appropriate for this case since ETS-4 particles ($d_{p}=0.7\times 10^{-6}$ m) are one order of magnitude lower than the inferior limit studied by the authors (range of $d_{p}=(6-450\times 10^{-6}$ m) [48].

For comparison, the fittings accomplished by the pseudo-first and pseudo-second order models are also presented, since they are two of the most adopted expressions in the literature for a huge variety of materials [12,50,51,52,53,54]. Figure 4 displays the evolution of the experimental (symbols) and calculated (lines) concentrations of Hg^2+^ in the solid phase along time, $\langle q_{A}\rangle$, and Table 3 contains the parameters fitted for each kinetic experiment, namely $k_{1}$ and $k_{2}$ (see Equations (S24) and (S25), in Appendix A). A good agreement between experimental data and model fitting was achieved, especially by the pseudo-second order model (AARD = 8.30%) being poorer for the pseudo-first order model (AARD = 14.50%). Nonetheless, in all cases the simple empirical models performed worse than the MS model (AARD = 5.81%).

For the systems using TPAOH solution to adjust the pH, the main deviations occurred in the transition between the steep ascendant part of each curve and the horizontal branch (Figure 4a’,b’), in particular for the first-order model, whereas the equilibrium values calculated by the pseudo-second order model matched accurately the experimental points (relative errors of 2.9% and 3.3%, respectively, for 13.6 and 15.6 g m^−3^ of ETS-4). For the system using NaOH, the major deviations occurred for the initial times (t<20 h) (Figure 4c’) and the model overestimated the equilibrium concentration of Hg^2+^ in the solid (relative error of 10.8%). Additionally, the pseudo-second order model kinetic constants, $k_{2}$, also confirmed that the ion exchange of Hg^2+^ is much faster in the presence of the TPA^+^ than Na^+^ ($k_{2}=\left(11.70\pm 1.30\right)\times 10^{-4}$
*versus*
$k_{2}=\left(0.56\pm 0.10\right)\times 10^{-4}$ m^3^ mol^−1^ h^−1^) when using the same amount of ETS-4 (ca. 15 g m^−3^) and initial concentration of mercury ($\mathrm{ca}.\;5\times 10^{-3}$ mol m^−3^).

## 5. Conclusions

The ion exchange of a hazardous metal ion, Hg^2+^, using the well-known titanosilicate no. 4 in sodium form (Na-ETS-4) as a cation exchanger was studied carrying out isothermal batch experiments and using two pH adjusting solutions with cations of different size (tetrapropylammonium and sodium cations). The results reveal that ETS-4 is highly effective and selective for Hg^2+^ uptake independently of the counter ions in solution. Moreover, the presence of Na^+^ in the fluid phase does not affect the equilibrium Hg^2+^ removal while it imparts a strong effect on the ion exchange kinetics (and thus on the equilibration time), since the addition of Na^+^ in the liquid phase decreases the driving force for sodium mass transfer.

In terms of modelling, equilibrium was accurately modeled by the mass action law and rigorously expressed in term of activities (Debye–Hückel model for solution and Wilson and NRTL models for the solid exchanger phase). Comparing to the ideal model, no major deviations from ideality were found in the range of concentrations studied, despite a noticeable biased increase at higher concentrations. Concerning kinetic modelling, the Maxwell–Stefan (MS) approach was adopted and the MS diffusivities of each pair of species and the film mass transfer coefficient were obtained. With an average deviation of 5.81%, the MS model described successfully the kinetic curves of mercury removal being notoriously better than the semi-empirical models frequently used in the literature, namely the pseudo-first (AARD = 15.08%) and pseudo-second (AARD = 10.78%) order equations. Nevertheless, the pseudo-second order model was able to describe the ion exchange of Hg^2+^ by the ETS-4 from the fluid phase.

## Figures and Tables

**Figure 1 materials-14-00011-f001:**
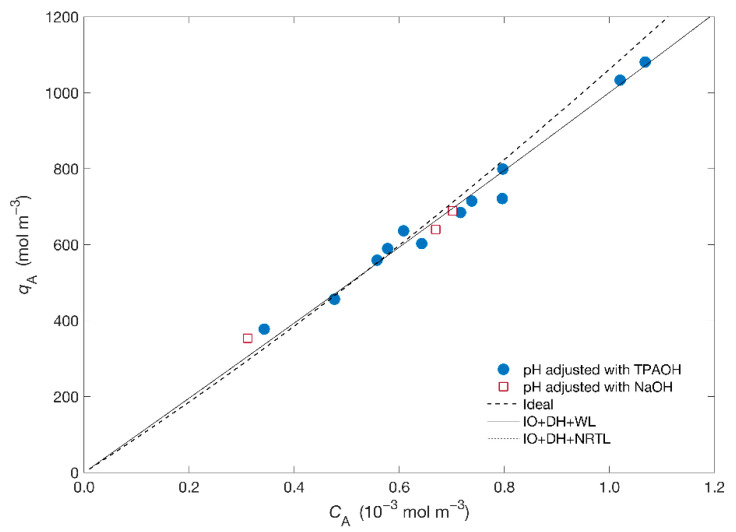
Equilibrium data (symbols) and isotherm modelling (lines) results for the system Hg^2+^/Na^+^/ETS-4. Acronyms: IO = Ioannidis et al. approach, DH = Debye–Hückel, WL = Wilson.

**Figure 2 materials-14-00011-f002:**
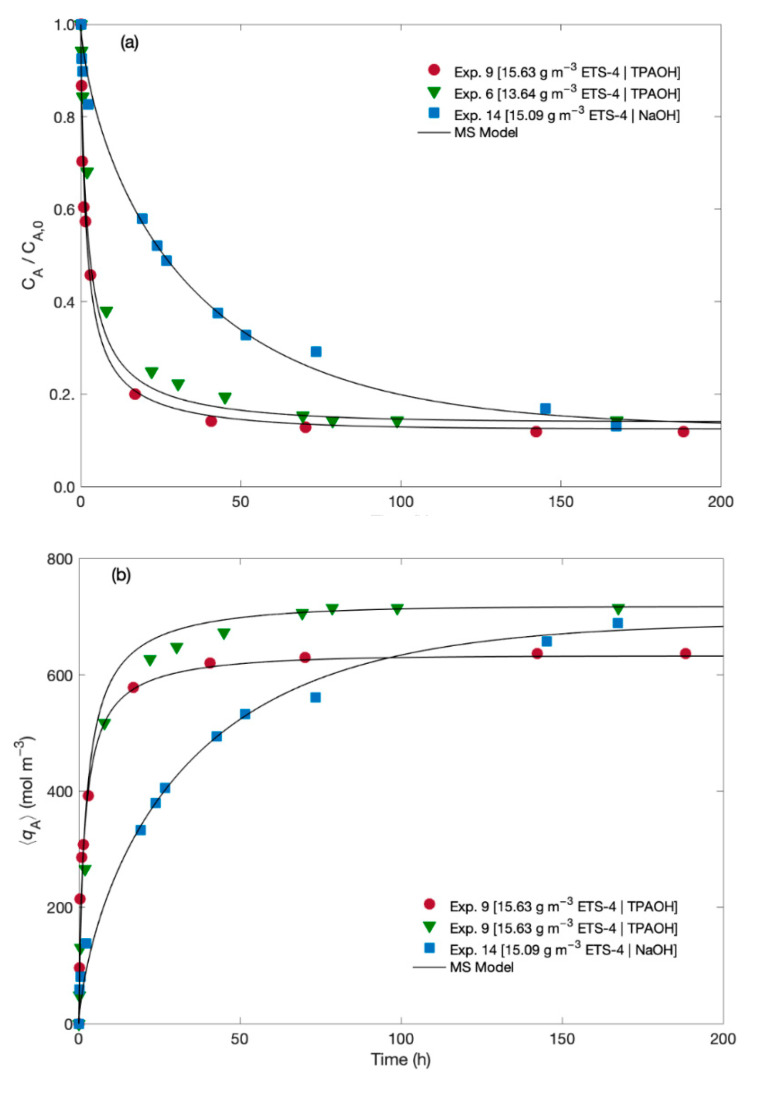
Experimental (symbols) and MS model (lines) results along time for (**a**) normalized Hg^2+^ concentration in the fluid, and (**b**) in the solid.

**Figure 3 materials-14-00011-f003:**
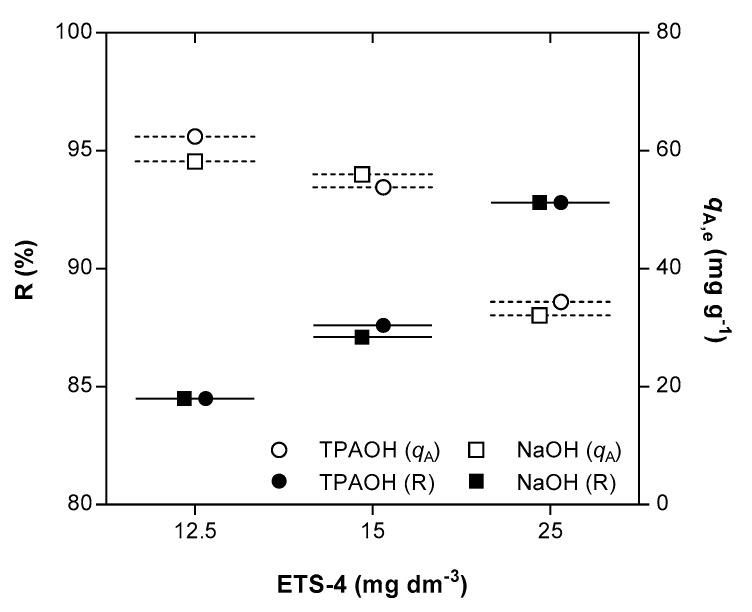
Removal (R) of Hg^2+^ (black symbols) and equilibrium concentrations of Hg^2+^ in the solid phase ($q_{A,e})$ (white symbols) as function of ETS-4 dose, when using TPAOH (circle) or NaOH (square) as pH adjusting solution. All data refers to the final equilibrium conditions attained.

**Figure 4 materials-14-00011-f004:**
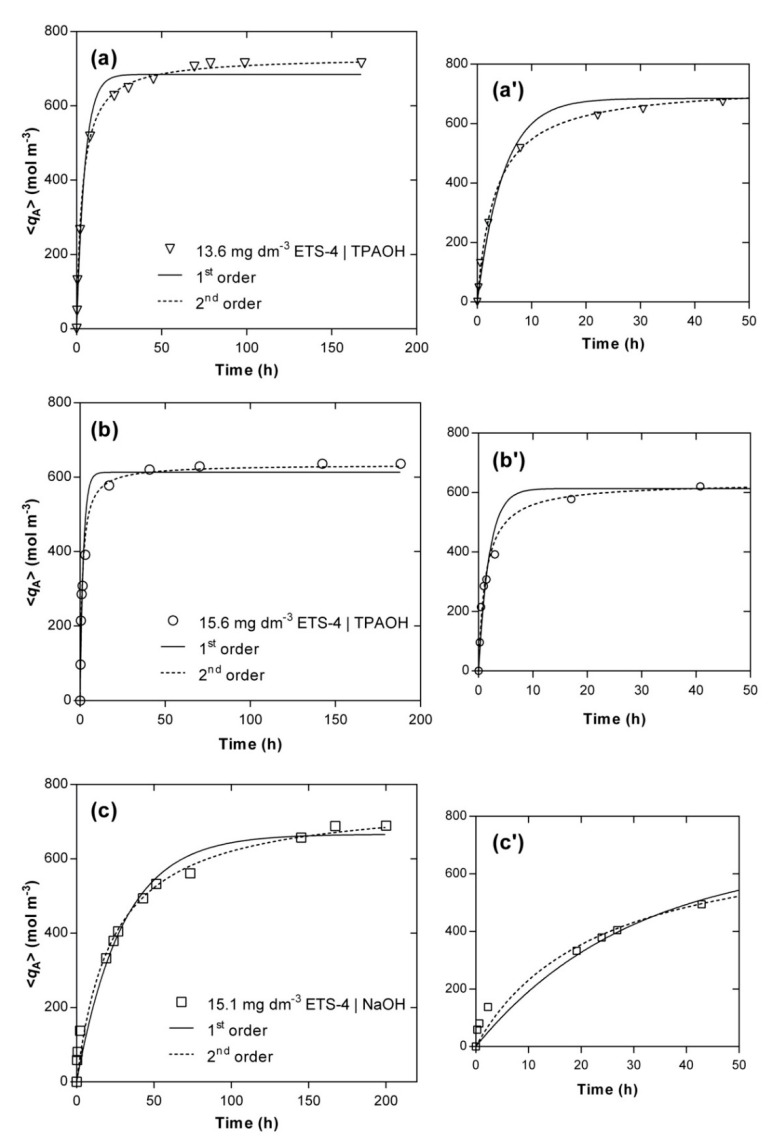
Experimental (symbols) and modelling (lines) results of Hg^2+^ concentration in the solid phase *versus* time. Experimental conditions: (**a**) TPAOH solution, 15.6 g m^−3^ of ETS-4 (Ο); (**b**) TPAOH solution, 13.6 g m^−3^ of ETS-4 (▽); (**c**) NaOH solution, 15.1 g m^−3^ of ETS-4 (☐). On the right side (**a’**–**c’**), the zoom-in for the first 50 h of each system is shown.

**Table 1 materials-14-00011-t001:** Features and initial conditions of each experiment.

Experiment No.	Experiment Type	Dose of ETS-4 (g m^−^^3^)	$C_{A,0}$ (10^−3^ mol m^−3^)	pH Adjust Solution
1	Equilibrium	7.15	4.58	TPAOH
2	Equilibrium	8.33	4.93	TPAOH
3	Equilibrium	11.10	4.83	TPAOH
4	Equilibrium	12.56	4.62	TPAOH
5	Equilibrium	12.62	4.93	TPAOH
6	Equilibrium & Kinetic	13.64	5.17	TPAOH
7	Equilibrium	14.52	4.62	TPAOH
8	Equilibrium	15.28	4.67	TPAOH
9	Equilibrium & Kinetic	15.63	5.13	TPAOH
10	Equilibrium	17.63	5.04	TPAOH
11	Equilibrium	20.23	4.67	TPAOH
12	Equilibrium	25.85	4.87	TPAOH
13	Equilibrium	12.55	4.31	NaOH
14	Equilibrium & Kinetic	15.09	5.43	NaOH
15	Equilibrium	25.19	4.34	NaOH

**Table 2 materials-14-00011-t002:** Equilibrium parameters obtained for the Hg^2+^/Na^+^/ETS-4 system. The range of ionic strength (*I*) in the liquid solution is $\left(6.74-6.83\right)\times 10^{-4}$ molal.

Model	$K_{B}^{A}$	Parameter 1 of ${\bar{\gamma }}_{i}$	Parameter 2 of ${\bar{\gamma }}_{i}$	AARD (%)
Ideal	$6.87\times 10^{-6}$	-	-	0.93
Debye–Hückel + Wilson	$4.43\times 10^{-6}$	0.4132	2.4199	0.89
Debye–Hückel + NRTL	$4.08\times 10^{-6}$	−2934	2667.1	0.89

**Table 3 materials-14-00011-t003:** Optimized parameters of Maxwell–Stefan, pseudo-first and pseudo-second order models, and respective calculated deviations (AARD).

Model	Maxwell–Stefan	Pseudo-first Order	Pseudo-Second Order
Parameters	*Ð*_ij_ (m^2^ s^−^^1^); *k*_f_ (m s^−^^1^)	*k*_1_ (10^−^^1^ h^−^^1^)	*k*_2_ (10^−^^4^ m^3^ mol^−^^1^ h^−^^1^)
$k_{f}$	5.09 × 10^−^^4^	*-*	*-*
TPAOH (Exp. 6 and Exp. 9)	*Ð*_AB_ = 4.33 × 10^−^^18^	*k*_1(Exp. 9)_ = 5.04 ± 0.72	*k*_2(Exp. 9)_ =11.70 ± 1.30
	*Ð*_AS_ = 4.25 × 10^−^^20^	*k*_1(Exp. 6)_ = 2.07 ± 0.26	*k*_2(Exp. 6)_ = 4.03 ± 0.04
	*Ð*_BS_ = 1.04 × 10^−^^20^		
NaOH (Exp. 14)	*Ð*_AB_ = 2.36 × 10^−^^16^	*k*_1(Exp. 14)_ = 0.34 ± 0.04	*k*_2(Exp. 14)_ = 0.56 ± 0.10
	*Ð*_AS_ = 3.09 × 10^−^^19^		
	*Ð*_BS_ = 5.03 × 10^−^^21^		
Global AARD (%)	5.81	14.50	8.30

**Table 4 materials-14-00011-t004:** Comparison between MS diffusivities and convective mass transfer coefficient optimized in this work and published in the literature.

System	Pore Diameter (Å)	${Ð}_{A,S}$ (m^2^s^−1^)	${Ð}_{B,S}$ (m^2^s^−1^)	${Ð}_{\mathrm{AB}}$ (m^2^s^−1^)	$k_{f}$ (m s^−1^)	Ref.
Hg^2+^/Na^+^/ETS-4 (*)	3–4	$2.57\times 10^{-18}$	$1.96\times 10^{-19}$	$6.24\times 10^{-20}$	$4.75\times 10^{-6}$	[49]
Cd^2+^/Na^+^/AV-6 (*)	8.34×6.17	$7.14\times 10^{-21}$	$5.98\times 10^{-19}$	$8.15\times 10^{-20}$	$1.26\times 10^{-4}$	[13]
Cd^2+^/Na^+^/ETS-4 (*)	3–4	$1.04\times 10^{-19}$	$9.07\times 10^{-19}$	$1.49\times 10^{-16}$	$6.10\times 10^{-4}$	[41]
Cd^2+^/Na^+^/ETS-10 (*)	4.9×7.6	$7.71\times 10^{-16}$	$9.18\times 10^{-15}$	$1.49\times 10^{-16}$	$1.01\times 10^{-4}$	[41]
Hg^2+^/Na^+^/ETS-4 (*)	3–4	$3.09\times 10^{-19}$	$5.03\times 10^{-21}$	$2.36\times 10^{-16}$	$5.09\times 10^{-4}$	This work
Hg^2+^/Na^+^/ETS-4 (**)	3–4	$4.25\times 10^{-20}$	$1.04\times 10^{-20}$	$4.33\times 10^{-18}$	$5.09\times 10^{-4}$	This work

* pH adjustment with NaOH; ** pH adjustment with TPAOH.

## Data Availability

No new data were created or analyzed in this study. Data sharing is not applicable to this article.

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
