# Peer review of "Mercury Removal from Aqueous Solution Using ETS-4 in the Presence of Cations of Distinct Sizes"

_materials, 2020, doi:10.3390/ma14010011_

Round 1

Reviewer 1 Report

The paper of Cardoso et al. analyse the equilibrium and kinetic aspects of inorganic mercury removal from aqueous solutions on ETS-4 as a function of the competing ions of variable sizes. This work is in line with the previous interest of the research group, adding valuable information with respect of the state of the art in the water depollution field.

The abstract is appropriate for the text’ content, while the title, even specific, seems too long (about 100 characters). The article is well constructed, the experiments well conducted and analysis well performed. The discussion is sound, as well as the modelling studies.

As minor points for revisions, I suggest the following:

Line 110: citation is needed for the ETS’ particle size, this parameter is varying as a function of the synthetic route

Line 113: mention the supplier for the TiCl3

Line 227, line 248 etc: in the present version of the ms, there are errors associated to the figures’ citations

Re-check the references list, I have noticed some inconsistences in the lists of authors.

Reviewer 2 Report

In the presented paper, the Authors provided experimental and theoretical investigation of mercury removal from solution using ETS-4. The study is designed and conducted correctly, but the scientific novelty of the results is not clear and should be highlighted. One can expect to observe the kinetic curves presented at Figs. 2 and 4. Well matching of these curves with parametric theoretical dependences is also expected. It would be interesting to read in the Abstract or Conclusions, what results were unexpected and attractive for readers. What are the features and main advantages of interaction of Hg with ESP-4 in comparison with the other ions / materials?

A couple of minor comments:

1) Figures numbering is wrong; equations numbering is possible wrong; constants k1 and k2 in Table 3 are not explained (may be they correspond to eqs. 18-19); 50-times variation of k2 value is seems to be strange (see Table 3).

2)  Table 4, why pore diameters are given as a multiplication of two numbers ("8.34×6.17")?

3) Kinetic constants and others fitting parameters derived from experimental dependences should be presented along with their dispersions.

Reviewer 3 Report

The authors investigated the use of a material for adsorption of mercury from aqueous solution. The reviewer suggests major revisions before the publication.

1) There are many mistakes. It is hard for the reviewer to check the manuscript due to the mistakes.

1-1) Check sentence in the introduction: "Isothermal batch experiments and two alkaline solutions were performed to measure equilibrium and kinetic data, were used as pH adjuster to assess competition effects,"

1-2) "Millipore membrane of 0.45×10−6 m" (Line 130) should be "Millipore membrane with pore size of 0.45×10−6 m (0.45 μm)"

1-3) "then analyzed" (Line 133) > analyze what? Mercury concentration ?

1-4) NAOH (Line 151) should be NaOH.

1-5) Comments for previous rejected version of the manuscript remain in Line 215-217. "This section may be divided by subheadings. It should provide a concise and precise description of the experimental results, their interpretation as well as the experimental conclusions that can be drawn."

1-6) Deleted figures from the previous version of the manuscript are cited in the current version (Line 227, 248) "Error! Reference source not found".

2) The definitions of the variables are not consistent. When they define cA and qA in equations (1), they denote concentrations and quantities of metals, although they sometimes indicate the concentrations and quantities of mercury. When the authors mentioned in Line 3.94×10−5 mol m-3 represents less than 0.8 % of the initial metal concentration (cA,0), ...", cA,0 means metal concentration, although the cA sometimes means mercury concentration. In addition the evidence (calculation procedure) for 0.8 % is not shown.

3) Improve the subheading " 4.1. Check the non exchangeability of tetrapropylammonium cation (TPA+) "

4) In figure 4, explain what are differences between (a) and (a'), (b) and (b'), (c) and (c'). It must be stated in the description for the figure, although the authors may have mentioned in the main text.

Reviewer 4 Report

In this paper, the authors studied the removal of the hazardous Hg2+ from aqueous solutions  by ion exchange 1using titanosilicate ETS-4. I really appreciate the correlation of exoerimental dat with modelling. I suggest this manuscript could be accepted for publications after minor revision.

1. I suggest the authors to improve the introduction adding more refernces to mehtod and materials used in literature for ions removal such as 

https://doi.org/10.3390/nano10061157

2. In section 4.2 the authors report the isotherm of the Hg2+/Na+/ETS-4 system expressed as molar concentration  of mercury. the experimental data were compared with models. More details on chosen models should be added.

3. Concerning kinetic modelling, the Maxwell-Stefan (MS) approach was adopted and the MS diffusivities of each pair of species and the film mass transfer coefficient were obtained. Also in this case, more details on this model should be added.

Reviewer 5 Report

The manuscript “Mercury removal from aqueous solution using ETS-4 in the presence of cations of distinct sizes: experimental and modelling studies of the ion exchange kinetics and equilibrium” presents routine studies concerning Hg2+ ion exchange using ETS-4 ion exchanger in the presence of TPA+ and Na+ as a competitive cations. The experiment is properly designed, properly conducted and the results are well-interpreted. Overall the manuscript is well-written. My only concern is a novelty of this paper and its impact on a progress in the science of materials. I suggest to highlight the aim of the conduction of presented studies.

Round 2

Reviewer 2 Report

All my previously submitted comments were satisfactory addressed. The revised version of the paper is much more clear than the initial variant. In my view, the manuscript should be accepted to Nanomaterials.

Reviewer 3 Report

The authors revised the manuscript. The paper can be published after minor revision.

1) Check the unit of ion strength (I) shown in the title of table 2 and on line 232. The reviewer feels something strange in the way authors use "molal" in the manuscript.

2) Check table 4 because the reviewer found no (**) in the table, although the authors mentioned both (*) and (**) in the footnote.

3) Some words are missing on lines 304-305. Please check again.  
